# Training Generative Adversarial Networks by Solving Ordinary Differential Equations

**Chongli Qin\*, Yan Wu\*, Jost Tobias Springenberg, Andrew Brock, Jeff Donahue, Timothy P. Lillicrap, Pushmeet Kohli**

DeepMind
chongliqin, yanwu@google.com

## Abstract

The instability of Generative Adversarial Network (GAN) training has frequently been attributed to gradient descent. Consequently, recent methods have aimed to tailor the models and training procedures to stabilise the discrete updates. In contrast, we study the continuous-time dynamics induced by GAN training. Both theory and toy experiments suggest that these dynamics are in fact surprisingly stable. From this perspective, we hypothesise that instabilities in training GANs arise from the integration error in discretising the continuous dynamics. We experimentally verify that well-known ODE solvers (such as Runge-Kutta) can stabilise training – when combined with a regulariser that controls the integration error. Our approach represents a radical departure from previous methods which typically use adaptive optimisation and stabilisation techniques that constrain the functional space (e.g. Spectral Normalisation). Evaluation on CIFAR-10 and ImageNet shows that our method outperforms several strong baselines, demonstrating its efficacy.

## 1 Introduction

The training of Generative Adversarial Networks (GANs) [11] has seen significant advances over the past several years. Most recently, GAN based methods have, for example, demonstrated the ability to generate images with high fidelity and realism such as the work of Brock et al. [3] and Karras et al. [15]. Despite this remarkable progress, there remain many questions regarding the instability of training GANs and their convergence properties.

In this work, we attempt to extend the understanding of GANs by offering a different perspective. We study the continuous-time dynamics induced by gradient descent on the GAN objective for commonly used losses. We find that under mild assumptions, the dynamics should converge in the vicinity of a differential Nash equilibrium, and that the rate of convergence is independent of the rotational part of the dynamics if we can follow the dynamics exactly. We thus hypothesise that *the instability in training GANs arises from discretisation of the continuous dynamics*, and we should focus on accurate integration to impart stability.

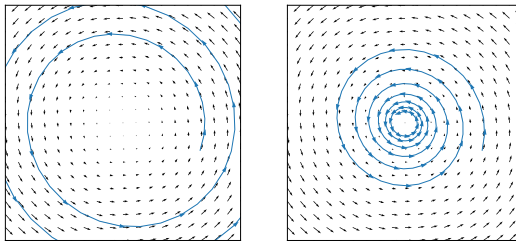

Figure 1: Left: divergence of integration with Euler's method – for a strong rotational field, step-size 0.01. Right: convergence with a second-order ODE solver (Heun's method) under the same step size and vector field. For details see Section 3.2.2.

Consistent with this hypothesis, we demonstrate that we can use standard methods for solving ordinary differential equations (ODEs) – such as Runge-Kutta – to solve for GAN parameters. In particular, we observe that more accurate time integration of the ODE yields better convergence as well as better

performance overall; a result that is perhaps surprising given that the integrators have to use noisy gradient estimates. We find that the main ingredient we need for stable GAN training is to avoid large integration errors, and a simple regulariser on the generator gradients is sufficient to achieve this. This alleviates the need for hard constraints on the functional space of the discriminator (e.g. spectral normalisation [21]) and enables GAN training without advanced optimisation techniques.

Overall, the contributions of this paper are as follows:

- We present a novel and practical view that frames GAN training as solving ODEs.
- We design a regulariser on the gradients to improve numerical integration of the ODE.
- We show that higher-order ODE solvers lead to better convergence for GANs. Surprisingly, our algorithm (ODE-GAN) can train GANs to competitive levels without any adaptive optimiser (e.g., Adam [16]) and explicit functional constraints (Spectral Normalisation).

## 2 Background and Notation

We study the GAN objective which is often described as a two-player min-max game with a *Nash equilibrium* at the saddle point, $\max_\theta \min_\phi \mathcal{J}(\theta, \phi)$, of the objective function

$$\mathcal{J}(\theta, \phi) = \mathbb{E}_{x \sim p(x)} \left[ \log(D(x; \theta)) \right] + \mathbb{E}_{z \sim p(z)} \left[ \log\left(1 - D(G(z; \phi); \theta)\right) \right], \tag{1}$$

where we denote the states of the discriminator and the generator respectively by their parameters $\theta$ and $\phi$. Following the convention of Goodfellow et al. [11], we use $x \sim p(x)$ to denote a data sample, and $\hat{x} = G(z; \theta)$ for the output of the generator (by transforming a sample from a noise source $z \sim p(z)$). Further, $\mathbb{E}_{x \sim p(x)} \left[ g(x) \right]$ stands for the expected value of a function $g(x)$ given the distribution $p(x)$. In practice, the problem is often transformed into one where the objective function is asymmetric (e.g., the generator's objective is changed to $\min_\phi \mathbb{E}_z \left[ -\log D(G(z; \phi), \theta) \right]$). We can describe this more general setting, which we focus on here, by using

$$\ell(\theta, \phi) = [\ell_D(\theta, \phi), \ \ell_G(\theta, \phi)], \tag{2}$$

to denote the loss vector of the discriminator-generator pair, and considering minimisation of $\ell_D(\theta, \phi)$ wrt. $\theta$ and $\ell_G(\theta, \phi)$ wrt. $\phi$. This setting is often referred to as a general-sum game. The original min-max problem from Eq. (1) can be captured in this notation by setting $\ell(\theta, \phi) = [-\mathcal{J}(\theta, \phi), \mathcal{J}(\theta, \phi)]$, and other commonly used variations (such as the non-saturating loss [11]) can also be accommodated.

## 3 Ordinary Differential Equations Induced by GANs

Here we derive the continuous form of GANs' training dynamics by considering the limit of infinitesimal time steps. Throughout, we consider a general-sum game with the loss pair $\ell(\theta, \phi) = [\ell_D(\theta, \phi), \ \ell_G(\theta, \phi)]$, as described in Section 2.

### 3.1 Continous Dynamics for GAN training

Given the losses from Eq. (2), the evolution of the parameters $(\theta, \phi)$, following simultaneous gradient descent (GD), is given by the following updates at iteration $k$

$$\theta_{k+1} = \theta_k - \alpha \, \frac{\partial \ell_D}{\partial \theta}(\theta_k, \phi_k) \, \Delta t$$

$$\phi_{k+1} = \phi_k - \beta \, \frac{\partial \ell_G}{\partial \phi}(\theta_k, \phi_k) \, \Delta t, \tag{3}$$

where $\alpha, \beta$ [1] are optional scaling factors. Then $\alpha \, \Delta t$ and $\beta \, \Delta t$ correspond to the learning rates in the discrete case. Previous work has analysed the dynamics in this discrete case (mostly focusing on the min-max game), see e.g., Mescheder et al. [19]. In contrast, we consider arbitrary loss pairs under the continuous dynamics induced by gradient descent. That is we consider $\theta(t)$ and $\phi(t)$ to have explicit dependence on continuous time. This perspective has been taken for min-max games in

Nagarajan and Kolter [22]. With this dependence $\theta_k = \theta(t_0 + k\Delta t)$ and $\phi_k = \phi(t_0 + k\Delta t)$. Then, as $\Delta t \to 0$, Eq. (3) yields a dynamical system described by the ordinary differential equation

$$\begin{pmatrix} \frac{d\theta}{dt} \\ \frac{d\phi}{dt} \end{pmatrix} = \mathbf{v}(\theta, \phi), \tag{4}$$

where $\mathbf{v} = -[\alpha \frac{\partial \ell_D}{\partial \theta}, \beta \frac{\partial \ell_G}{\partial \phi}]$. This is also known as infinitesimal gradient descent [26].

Perhaps surprisingly, if we view the problem of training GANs from this perspective we can make the following observation. *Assuming we track the dynamical system exactly – and the gradient vector field* $\mathbf{v}$ *is bounded – then in the vicinity of a differential Nash equilibrium[2]* $(\theta_c, \phi_c)$*,* $(\theta, \phi)$ *converges to this point at a rate independent of the frequency of rotation with respect to the vector field [3]*.

There are two direct consequences from this observation: First, changing the rate of rotation of the vector field does not change the rate of convergence. Second, if the velocity field has no subspace which is purely rotational, then it suggests that in principle GAN training can be reduced to the problem of accurate time integration. Thus if the dynamics are attracted towards a differential Nash they should converge (though reaching this regime depends on initial conditions).

## 3.2 Convergence of GAN Training under Continous Dynamics

We next show that, under mild assumptions, close to a differential Nash equilibrium, the continuous time dynamics from Eq (4) converge to the equilibrium point for GANs under commonly used losses. We further clarify that they locally converge even under a strong rotational field as in Fig. 1, a problem studied in recent GAN literature (see also e.g. Balduzzi et al. [2], Gemp and Mahadevan [10] and Mescheder et al. [19]).

### 3.2.1 Analysis of Linearised Dynamics for GANs in the Vicinity of Nash Equilibria

We here show that, *in the vicinity of a differential Nash equilibrium*, the dynamics converge unless the field has a purely rotational subspace. We prove convergence under this assumption for the continuous dynamics induced by GANs using the cross-entropy or the non-saturated loss. This analysis has strong connections to work by Nagarajan and Kolter [22], where local convergence was analysed in a more restricted setting for min-max games.

Let us consider the dynamics close to a local Nash equilibrium $(\theta^*, \phi^*)$ where $\mathbf{v}(\theta^*, \phi^*) = 0$ by definition. We denote the increment as $\delta = [\theta, \phi] - [\theta^*, \phi^*]$, then $\dot{\delta} = -H\delta + O(|\delta|^2)$ where $H = -[\frac{\partial \mathbf{v}}{\partial \theta}, \frac{\partial \mathbf{v}}{\partial \phi}]$. This Jacobian has the following form for GANs:

$$H = \begin{pmatrix} \frac{\partial^2 \ell_D}{\partial \theta^2} & \frac{\partial^2 \ell_D}{\partial \phi \partial \theta} \\ \frac{\partial^2 \ell_G}{\partial \theta \partial \phi} & \frac{\partial^2 \ell_G}{\partial \phi^2} \end{pmatrix}\Bigg|_{(\theta^*, \phi^*)}. \tag{5}$$

For the cross-entropy loss, the non-saturated loss, and the Wasserstein loss we observe that at the local Nash, $\frac{\partial^2 \ell_D}{\partial \phi \partial \theta} = -\frac{\partial^2 \ell_G}{\partial \theta \partial \phi}^T$ (please see Appendix B for a derivation). Consequently, for GANs, we consider the linearised dynamics of the following form:

$$\begin{pmatrix} \frac{d\theta}{dt} \\ \frac{d\phi}{dt} \end{pmatrix} = -\begin{pmatrix} A & B^T \\ -B & C \end{pmatrix} \begin{pmatrix} \theta \\ \phi \end{pmatrix}, \tag{6}$$

where $A, B, C$ denote the elements of $H$.

**Lemma 3.1.** *Given a linearised vector field of the form shown in Eq. (6) where either* $A \succ 0$ *and* $C \succeq 0$ *or* $A \succeq 0$ *and* $C \succ 0$*; and B is full rank. Following the dynamics will always converge to* $\mathbf{v} = [0, 0]^T$.

*Proof.* The proof requires three parts: We first show that $H$ of this form must be invertible in Lemma A.1, then we show, as a result, in Lemma A.2, that the real part of the eigenvalues for $H$ must be strictly greater than 0. The third part follows from the previous Lemmas: as the eigenvalues of $H$, $\{u_0 + iv_0, \cdots, u_n + iv_n\}$, satisfy $u_i > 0$ the solution of the linear system converges at least at rate $e^{-\min_i(u_i)t}$. Consequently, as $t \to \infty$, $\mathbf{v}(t) \to [0, 0]^T$. □

We observe for the cross entropy loss and the non-saturating loss $A = \partial^2 \ell_D / \partial \theta^2 \succeq 0$, if we train with piece-wise linear activation functions – for details see Appendix C. Thus for these losses and if convergence occurs (i.e. under the assumption that one of $A$ or $C$ is positive definite, the other semi-positive definite), we should converge to the differential Nash equilibrium in its vicinity if we follow the dynamics exactly.

**Remark.** *We note that if we consider a non-hyperbolic fixed point, namely the eigenvalues of the Jacobian of the dynamics have no real-parts, then following the dynamics will only rotate around the fixed point. An example with such a non-hyperbolic fixed point is given by $\ell_D(\theta, \phi) = \theta F \phi$, $\ell_G(\theta, \phi) = -\theta F \phi$, for any $F \neq 0$. However, we argue that this special case is not applicable for neural network based GANs. To see this, note that the discriminator loss on the real examples is totally independent of the generator. Consequently, in this case we should consider $\ell_D(\theta, \phi) = f(\theta) + \theta F \phi$.*

### 3.2.2 Convergence Under a Strong Rotational Field

As an illustrative example of Lemma 3.1, and to verify the importance of accurate time integration of the dynamics, we provide a simple toy example with a strong rotational vector field. Consider a two-player game where the loss functions are given by:

$$\ell_D(\theta, \phi) = \frac{1}{2}\epsilon\theta^2 - \theta\phi, \quad \ell_G(\theta, \phi) = \theta\phi,$$

This game maps directly to the linearised form from above (and in addition it satisfies $\partial^2 \ell_D / \partial \theta^2 \succ 0$ for $\epsilon > 0$, as in Lemma 3.1). When $\epsilon > 0$, the Nash is at $[0, 0]^T$. The vector field of the system is

$$\begin{pmatrix} \frac{\mathrm{d}\theta}{\mathrm{d}t} \\ \frac{\mathrm{d}\phi}{\mathrm{d}t} \end{pmatrix} = - \begin{pmatrix} \epsilon & -1 \\ 1 & 0 \end{pmatrix} \begin{pmatrix} \theta \\ \phi \end{pmatrix}. \tag{7}$$

When $\epsilon < 2$, it has the analytical solution

$$\theta(t) = e^{-\epsilon t/2}(a_0 \cos(\omega t) + b_0 \sin(\omega t)), \tag{8}$$

$$\phi(t) = e^{-\epsilon t/2}(a_1 \cos(\omega t) + b_1 \sin(\omega t)), \tag{9}$$

where $\omega = \sqrt{4 - \epsilon^2}/2$ and $a_0, a_1, b_0, b_1$ can be determined from initial conditions. Thus the dynamics will converge to the Nash as $t \to \infty$ independent of the initial conditions. In Fig. 1 we compare the difference of using a first order numerical integrator (Euler's) vs a second order integrator (two-stage Runge-Kutta, also known as Heun's method) for solving the dynamical system numerically when $\epsilon = 0.1$. When we chose $\Delta t = 0.2$ for 200 timesteps, Euler's method diverges while RK2 converges.

## 4 ODE-GANs

In this section we outline our practical algorithm for applying standard ODE solvers to GAN training; we call the resulting model an ODE-GAN. With few exceptions (such as the exponential function), most ODEs cannot be solved in closed-form and instead rely on numerical solvers that integrate the ODEs at discrete steps. To accomplish this, an ODE solver approximates the ODE's solution as a cumulative sum in small increments.

We denote the following to be an update using an ODE solver; which takes the velocity function $\mathbf{v}$, current parameter states $(\theta_k, \phi_k)$ and a step-size $h$ as input:

$$\begin{pmatrix} \theta_{k+1} \\ \phi_{k+1} \end{pmatrix} = \texttt{ODEStep}(\theta_k, \phi_k, \mathbf{v}, h). \tag{10}$$

Note that $\mathbf{v}$ is the function for the velocity field defined in Eq. (4). For an Euler integrator the `ODEStep` method would simply compute $[\phi_k, \theta_k]^T + h\mathbf{v}(\phi, \theta)$, and is equivalent to simultaneous gradient descent with step-size $h$. However, as we will outline below, higher-order methods such as the fourth-order Runge-Kutta method can also be used. After the update step is computed, we add a small amount of regularisation to further control the truncation error of the numerical integrator. A listing of the full procedure is given in Algorithm 1.

### 4.1 Numerical Solvers for ODEs

We consider several classical ODE solvers for the experiments in this paper, although any ODE solver may be used with our method.

---
**Algorithm 1** Training an ODE-GAN
---
**Require:** Initial states $(\theta, \phi)$, step size $h$, velocity function $\mathbf{v}(\theta, \phi) = -[\frac{\partial \ell_D}{\partial \theta}, \frac{\partial \ell_G}{\partial \phi}]$, regularization multiplier $\lambda$, and initial step counter $i = 0$, maximum iteration $I$
  **if** $i < I$ **then**
    $g_\theta \leftarrow \nabla_\theta \parallel \frac{\partial \ell_G}{\partial \phi}|_{(\theta, \phi)} \parallel^2$
    $\theta, \phi \leftarrow \texttt{ODEStep}(\theta, \phi, \mathbf{v}, h)$
    $\theta \leftarrow \theta - h\lambda g_\theta$
    $i \leftarrow i + 1$
  **end if**
  **return** $(\theta, \phi)$
---

**Different Orders of Numerical Integration**
We experimented with a range of solvers with different orders. The order controls how the truncation error changes with respect to step size $h$. Namely, the errors of first order methods reduce linearly with decreasing $h$, while the errors of second order methods decrease quadratically. The ODE solvers considered in this paper are: first order Euler's method, a second order Runge Kutta method – Heun's method – (RK2), and fourth order Runge-Kutta method (RK4). For details on the explicit updates to the GAN parameters applied by each of the methods we refer to Appendix E. Computational costs for calculating `ODEStep` grow with higher-order integrators; in our implementation, the most expensive solver considered (RK4) was less than $2\times$ slower (in wall-clock time) than standard GAN training.

**Connections to Existing Methods for Stabilizing GANs** We further observe (derivation in Appendix F) that existing methods such as Consensus Optimization [19], Extragradient [17, 5] and Symplectic Gradient Adjustment [2] can be seen as approximating higher order integrators.

## 4.2 Practical Considerations for Stable integration of GANs

To guarantee stable integration, two issues need to be considered: exploding gradients and the noise from mini-batch gradient estimation.

**Gradient Regularisation.** When using deep neural networks in the GAN setting, gradients are not *a priori* guaranteed to be bounded (as required for ODE integration). In particular, looking at the form of $\mathbf{v}(\theta, \phi)$ we can make two observations. First, the discriminator's gradient is grounded by real data and we found it to not explode in practice. Second, the generator's gradient $-\frac{\partial \ell_G}{\partial \phi}$ can explode easily, depending on the discriminator's functional form during learning[4]. This prevents us from using even moderate step-sizes for integration. However, in contrast to solving a standard ODE, the dynamics in our ODE are given by learned functions. Thus we can control the gradient magnitude to some extent. To this end, we found that a simple mitigation is to regularise the discriminator such that the gradients are bounded. In particular we use the regulariser

$$R(\theta) = \lambda \left\| \frac{\partial \ell_G}{\partial \phi} \right\|^2 , \tag{11}$$

whose gradient $\nabla_\theta R(\theta)$ wrt. the discriminator parameters $\theta$ is well-defined in the GAN setting. Importantly, since this regulariser vanishes as we approach a differential Nash equilibrium, it does not change the parameters at the equilibrium. Empirically, we found that this regulariser can efficiently control integration errors (see Appendix D for details).

This regulariser is similar to the one proposed by Nagarajan and Kolter [22]. While they suggest adjusting updates to the generator (using the regularizer $(\partial \ell_D / \partial \theta)^2$), we found this did not control the gradient norm well; as our experiments suggest that it is the gradient of the loss wrt. the generator parameters that explodes in practice. It also shares similarities with the penalty term from Gulrajani et al. [13]. However, we regularise the gradient with respect to the parameters rather than the input.

**A Note on Using Approximate Gradients.** Practically, when numerically integrating the GAN equations, we resort to Monte-Carlo approximation of the expectations in $\mathcal{J}(\phi, \theta)$. As is common, we use a mini-batch of $N$ samples from a fixed dataset to approximate expectations involving $p(x)$ and use the same number of random samples from $p(z)$ to approximate respective expectations. Calculating the partial derivatives $\frac{\partial \ell_D}{\partial \theta}, \frac{\partial \ell_G}{\partial \phi}$ based on these Monte-Carlo estimates leads to approximate gradients, i.e. we use $\tilde{v}(\theta, \phi) \approx v(\theta, \phi) + \zeta$, where $\zeta$ approximates the noise introduced due to sampling. We note that, the continuous GAN dynamics themselves are not stochastic (i.e. they do not form a

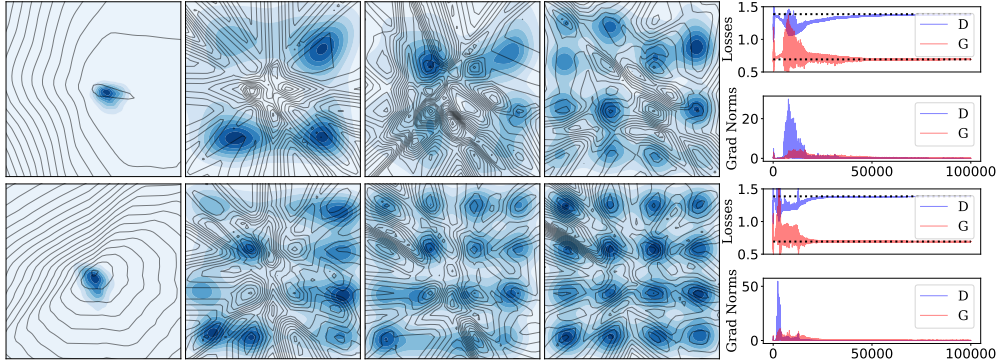

Figure 2: Using Euler (top row) and RK4 (second row) with $\lambda = 0.07$, $h = 0.03$. We show the progression at 0, 6k, 12k and 18k steps. The black contours show the discriminator function, blue heat maps indicate the generator density. The right figure shows the losses and the norms of the gradients. The dashed lines indicate the Nash equilibrium ($\log(4)$ for discriminator, $\log(2)$ for generator).

stochastic differential equation) and noise purely arises from approximation errors, which decrease with more samples. While one could expect the noise to affect integration quality we empirically observed competitive results with integrators of order greater than one.

## 5 Experiments

We evaluate ODE-GANs on data from a mixture of Gaussians, CIFAR-10 [18] (unconditional), and ImageNet [7] (conditional). All experiments use the non-saturated loss $\ell_\phi = \mathbb{E}_z \left[ -\log D(G(z)) \right]$, which is also covered by our theory. Remarkably, our experiments suggest that simple Runge-Kutta integration with the regularisation is competitive (in terms of FID and IS) with common methods for training GANs, while simultaneously keeping the discriminator and generator loss close to the true Nash payoff values at convergence – of $\log(4)$ for the discriminator and $\log(2)$ for the generator. Experiments are performed *without* Adam [16] unless otherwise specified and we evaluate IS and FID on 50k images. We also emphasise inspection of the behaviour of training at convergence *and not just at the best scores over training* to determine whether a stable fixed point is found. Code is available at `https://github.com/deepmind/deepmind-research/tree/master/ode_gan`.

### 5.1 Mixture of Gaussians

We compare training with different integrators for a mixture of Gaussians (Fig. 2). We use a two layer MLP (25 units) with ReLU activations, latent dimension of 32 and batch size of 512. This low dimensional problem is solvable both with Euler's method as well as Runge-Kutta with gradient regularisation – but without regularisation gradients grow large and integration is harmed, see Appendix H.3. In Fig. 2, we see that both the discriminator and generator converge to the Nash payoff value as shown by their corresponding losses; at the same time, all the modes were recovered by both players. As expected, the convergence rate using Euler's method is slower than RK4.

### 5.2 CIFAR-10 and ImageNet

We use 50K samples to evaluate IS/FID. Unless otherwise specified, we use the DCGAN from [23] for the CIFAR-10 dataset and ResNet-based GANs [13] for ImageNet.

#### 5.2.1 Different Orders of ODE Solvers

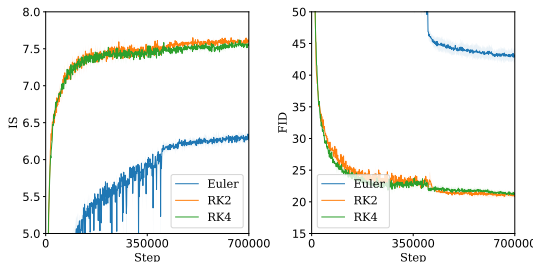

Figure 3: Comparison between different orders of integrators using $\lambda = 0.002$ and $h = 0.02$.

As shown in Fig. 3, we find that moving from a first order to a second order integrator can significantly improve training convergence. But when we go past second order we see diminishing returns. We further observe that higher order methods allow for much larger step sizes: Euler's method becomes unstable with $h \geq 0.04$ while Heun's method (RK2) and RK4 do not. On the other hand, if we increase the

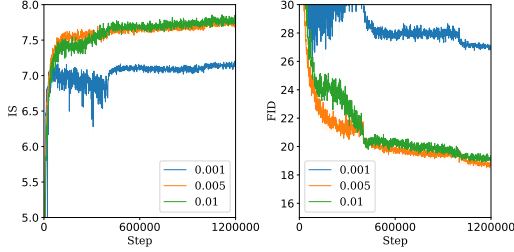
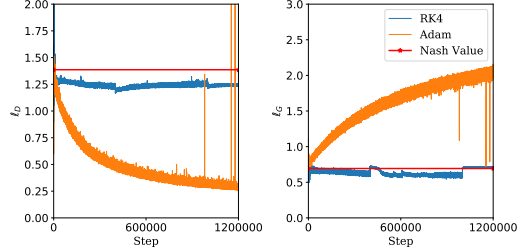

Figure 4: Comparison of runs with different regularisation weight $\lambda$ shown in legend. The step size used is $h = 0.04$, all with RK4 integrator.

Figure 5: The evolution of loss values for discriminator (left) and generator (right) for ODE-GAN (RK4) versus training with Adam optimisation.

regularisation weight, the performance gap between Euler and RK2 is reduced (results for higher regularisation $\lambda = 0.01$ tabulated in Table H2 in the appendix). This hints at an implicit effect of the regulariser on the truncation error of the integrator.

### 5.2.2 Effects of Gradient Regularisation

The regulariser controls the truncation error by penalising large gradient magnitudes (see Appendix D). We illustrate this with an embedded method (Fehlberg method, comparing errors between 3rd and 2nd order), which tracks the integration error over the course of training. We observe that larger $\lambda$ leads to smaller error. For example, $\lambda = 0.04$ gives an average error of $0.0003$, $\lambda = 0.01$ yields $0.0015$ with RK4 (see appendix Fig. H4). We depict the effects of using different $\lambda$ values in Fig. 4.

### 5.2.3 Loss Profiles for the Discriminator and Generator

Our experiments reveal that using more accurate ODE solvers results in loss profiles that differ significantly to curves observed in standard GAN training, as shown in Fig. 5. Strikingly, we find that the discriminator loss and the generator loss stay very close to the values of a Nash equilibrium, which are $\log(4)$ for the discriminator and $\log(2)$ for the generator (shown by red lines in Fig. 5, see [11]). In contrast, the discriminator dominates the game when using the Adam optimiser, evidenced by a continuously decreasing discriminator loss, while the generator loss increases during training. This imbalance correlates with the well-known phenomenon of worsening FID and IS in late stages of training (we show this in Table H2, see also e.g. Arjovsky and Bottou [1]).

### 5.2.4 Comparison to Standard GAN training

This section compares using ODE solvers versus standard methods for GAN training. Our results challenge the widely-held view that adaptive optimisers are necessary for training GANs, as revealed in Fig. 7. Moreover, the often observed degrading performance towards the end of training disappears with improved integration. To our knowledge, this is the first time that competitive results for GAN training have been demonstrated for image generation without adaptive optimisers. We also compare ODE-GAN with SN-GAN in Fig. 6 (re-trained and tuned using our code for fair comparison) and find that ODE-GAN can improve significantly upon SN-GAN for both IS and FID. Comparisons to more baselines are listed in Table 1. For the DCGAN architecture, ODE-GAN (RK4) achieves 17.66

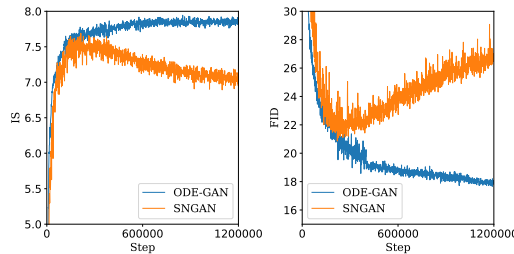
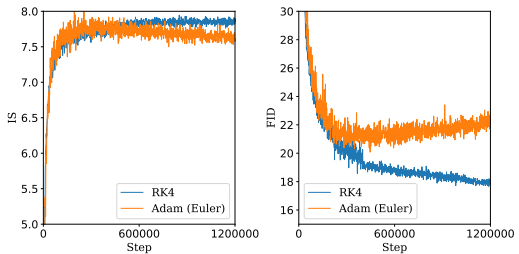

Figure 6: Here we compare ODE-GAN (with RK4 as ODEStep and $\lambda = 0.01$) to SN-GAN.

Figure 7: Here we compare using the convergence of RK4 to using Adam, $\lambda = 0.01$ for both.

Table 1: Numbers taken from the literature are cited. ‡ denotes reproduction in our code. "Best" and "final" indicate the best scores and scores at the end of 1 million update steps respectively. The means and standard deviations (shown by ±) are computed from 3 runs with different random seeds. We use **bold face** for the best scores across each category incl. those within one standard deviation.

| Method | FID (best) / FID(final) | IS (best)/IS(final) |
|---|---|---|
| **CIFAR-10 Unconditional** | | |
| **– DCGAN –** | | |
| ODE-GAN(RK4) | **17.66 ± 0.38 / 18.05 ± 0.53** | **7.97 ± 0.03 / 7.86 ± 0.09** |
| ODE-GAN(RK4+Adam) | **17.47 ± 0.30** / 23.20 ± 0.95 | **8.00 ± 0.06** / 7.59 ± 0.14 |
| SN-GAN‡ | 21.71 ± 0.61 / 26.16 ± 0.27 | 7.60 ± 0.06 / 7.02 ± 0.02 |
| **– ResNet –** | | |
| ODE-GAN (RK4) | **11.85 ± 0.21 / 12.50 ± 0.30** | **8.61 ± 0.06 / 8.51 ± 0.01** |
| ODE-GAN (RK4 + Adam) | 12.11 ± 0.28 / 20.32 ± 1.17 | 8.23 ± 0.04 / 7.92 ± 0.03 |
| SN-GAN‡ | 15.97 ± 0.22 / 23.98 ± 2.08 | 7.71 ± 0.05 / 7.20 ± 0.22 |
| WGAN-ALP (ResNet) [27] | 12.96 ± 0.35 / — | 8.56 / — |
| **– Evaluated with 5k samples –** | | |
| SN-GAN (DCGAN) [21] | 29.3 / — | 7.42 ± 0.08 / — |
| WGAN-GP (DCGAN) [13] | 40.2 / — | 6.68 ± 0.06 / — |
| SN-GAN (ResNet) [21] | 21.7 ± 0.21 / — | 8.22 ± 0.05 / — |
| **ImageNet $128 \times 128$ Conditional** | | |
| **– ResNet –** | | |
| ODE-GAN (RK4) | **26.16 ± 0.75 / 28.42 ± 1.46** | **38.71 ± 0.82 / 36.54 ± 1.53** |
| SN-GAN‡ | 37.05 ± 0.26 / 41.07 ± 0.46 | 31.52 ± 0.25 / 29.16 ± 0.20 |

FID as well as 7.97 IS, note that these best scores are remarkably close to scores we observe at the end of training.

ODE solvers and adaptive optimisers can also be combined. We considered this via the following approach (listed as ODE-GAN(RK4+Adam) in tables): we use the adaptive learning rates computed by Adam to scale the gradients used in the ODE solver. Table 1 shows that this combination reaches similar best IS/FID, but then deteriorates. This observation suggests that Adam can efficiently accelerate training, but the convergence properties may be lost due to the modified gradients – analysing these interactions further is an interesting avenue for future work. For a more detailed comparison of RK4 and Adam, see Table H2 in the appendix.

In Tables 1 and H1, we present results to test ODE-GANs at a larger scale. For CIFAR-10, we experiment with the ResNet architecture from Gulrajani et al. [13] and report the results for baselines using the same model architecture. ODE-GAN achieves 11.85 in FID and 8.61 in IS. Consistent with the behavior we see in DCGAN, ODE-GAN (RK4) results in stable performance throughout training. Additionally, we trained a conditional model on ImageNet $128 \times 128$ with the ResNet used in SNGAN without Spectral Normalisation (for further details see Appendix G and H.1). ODE-GAN achieves 26.16 in FID and 38.71 in IS. We have also trained a larger ResNet on ImageNet (see Appendix G), where we can obtain 22.29 for FID and 46.17 for IS using ODE-GAN (see Table H1). Consistent with all previous experiments, we find that the performance is stable and does not degrade over the course of training: see Fig. H1 and Tables 1 and H1.

## 6 Discussion and Relation to Existing Work

Our work explores higher-order approximations of the continuous dynamics induced by GAN training. We show that improved convergence and stability can be achieved by faithfully following the vector field of the adversarial game – without the necessity for more involved techniques to stabilise training, such as those considered in Salimans et al. [25], Balduzzi et al. [2], Gulrajani et al. [13], Miyato et al. [21]. Our empirical results thus support the hypothesis that, at least locally, the GAN game is not inherently unstable. Rather, *the discretisation of GANs' continuous dynamics, yielding inaccurate time integration, causes instability.* On the empirical side, we demonstrated for training on CIFAR-10 and ImageNet that both Adam [16] and spectral normalisation [21], two of the most popular techniques, may harm convergence, and that they are not necessary when higher-order ODE solvers are available.

The dynamical systems perspective has been employed for analysing GANs in previous works [22, 28, 19, 2, 10]. They mainly consider simultaneous gradient descent to analyse the discretised dynamics. In contrast, we study the link between GANs and their underlying continuous time dynamics which prompts us to use higher-order integrators in our experiments. Others made related connections: for example, using a second order ODE integrator was also considered in a simple 1-D case for GANs in Gemp and Mahadevan [10], and Nagarajan and Kolter [22] also analysed the continuous dynamics in a more restrictive setting – in a min-max game around the optimal solution. We hope that our paper can encourage more work in the direction of this connection [28], and adds to the valuable body of work on analysing GAN training convergence [29, 9, 20].

Lastly, it is worth noting that viewing traditionally discrete systems through the lens of continuous dynamics has recently attracted attention in other parts of machine learning. For example, the Neural ODE [6] interprets the layered processing of residual neural networks [14] as Euler integration of a continuous system. Similarly, we hope our work can contribute towards establishing a bridge for utilising tools from dynamical systems for generative modelling.

## Broader Impact

This work offers a perspective on training generative adversarial networks through the lens of solving an ordinary differential equation. As such it helps us connect an important part of current studies in machine learning (generative modelling) to an old and well studied field of research (integration of dynamical systems).

Making this connection more rigorous over time could help us understand how to better model natural phenomena, see e.g. Zoufal et al. [30] and Casert et al. [4] for recent steps in this direction. Further, tools developed for the analysis of dynamical systems could potentially help reveal in what form exploitable patterns exist in the models we are developing – or their dynamics – and as a result contribute to the goal of learning robust and fair representations [12].

The techniques proposed in this paper make training of GAN models more stable. This may result in making it easier for non-experts to train such models for beneficial applications like creating realistic images or audio for assistive technologies (e.g. for the speech-impaired, or technology for restoration of historic text sources). On the other hand, the technique could also be used to train models used for nefarious applications, such as forging images and videos (often colloquially referred to as "DeepFakes"). There are some research projects to find ways to mitigate this issue, one example is the DeepFakes Detection Challenge [8].

## Acknowledgments and Disclosure of Funding

We would especially like to thank David Balduzzi for insightful initial discussions and Ian Gemp for careful reading of our paper and feedback on the work.

## Footnotes

[1] For brevity, we set $\alpha = \beta = 1$ in our analysis.

[2]We refer to Ratliff et al. [24] for an overview on local and differential Nash equilibria.

[3]Namely, the magnitude of the imaginary eigenvalues of the Hessian does not affect the rate of convergence.

[4]We have observed large gradients when the generator samples are poor, where the discriminator may perfectly distinguish the generated samples from data. This potentially sharp decision boundary can drive the magnitude of the generator' gradients to infinity.

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
