[Supplementary Material]

# Supplementary: Training Generative Adversarial Networks by Solving Ordinary Differential Equations

**Chongli Qin\*, Yan Wu\*, Jost Tobias Springenberg, Andrew Brock, Jeff Donahue**
**Timothy P. Lillicrap, Pushmeet Kohli**
DeepMind
chongliqin, yanwu@google.com

## Abstract

We present details on the proofs from the main paper in Sections A-C. We include analysis on the effects of regularisation on the truncation error (Section D). Update rules for the ODE solvers considered in the main paper are presented in Section E. The connections between our method and Consensus optimisation, SGA and extragradient are reported in Section F. Further details of experiments/additional experimental results are in Section G-H. Image samples are shown in Section I.

## A  Real Part of the Eigenvalues are Positive

**Definition A.1.** *The strategy $(\theta^*, \phi^*)$ is a differential Nash equilibrium if at this point the first order derivatives and second order derivatives satisfy $\partial \ell_D / \partial \theta = \partial \ell_G / \partial \phi = 0$ and $\partial^2 \ell_D / \partial \theta^2 \succ 0$, $\partial^2 \ell_G / \partial \phi^2 \succ 0$ (see Ratliff et al. [12]).*

**Lemma A.1.** *For a matrix of form*

$$H = \begin{pmatrix} A & B^T \\ -B & C \end{pmatrix}.$$

*Given $B$ is full rank, if either $A \succ 0$ and $C \succeq 0$ or $A \succeq 0$ and $C \succ 0$, $H$ is invertible.*

*Proof.* Let's assume that $A \succ 0$, then we note $C + BA^{-1}B^T \succ 0$. Thus by definition $A$ and $C + BA^{-1}B^T$ are invertible as there are no zero eigenvalues. The Schur complement decomposition of $H$ is given by

$$H = \begin{pmatrix} I & 0 \\ -BA^{-1} & I \end{pmatrix} \begin{pmatrix} A & 0 \\ 0 & C + BA^{-1}B^T \end{pmatrix} \begin{pmatrix} I & A^{-1}B^T \\ 0 & I \end{pmatrix}$$

Each matrix in this is invertible, thus $H$ is invertible. □

**Lemma A.2.** *For a matrix of form*

$$\begin{pmatrix} A & B^T \\ -B & C \end{pmatrix},$$

*if either $A \succ 0$ and $C \succeq 0$ or vice versa, and $B$ is full rank, the positive part of the eigenvalues of $H$ is strictly positive.*

*Proof.* We assume $A \succ 0$ and $C \succeq 0$. The proof for $A \succeq 0$ and $C \succ 0$ is analogous.

To see that the real part of the eigenvalues are positive, we first note that the matrix satisfies the following:

$$H = \begin{pmatrix} A & B^T \\ -B & C \end{pmatrix} \Rightarrow \mathbf{x}^T H \mathbf{x} \geq 0.$$

To see this more explicitly, note

$$[\mathbf{x}^T, \mathbf{y}^T] \, H \, [\mathbf{x}, \mathbf{y}] = [\mathbf{x}^T, \mathbf{y}^T] \begin{pmatrix} A & B^T \\ -B & C \end{pmatrix} [\mathbf{x}, \mathbf{y}] = \mathbf{x}^T A \mathbf{x} + \mathbf{y}^T C \mathbf{y} \geq 0 \; \forall \mathbf{x} \in \mathbb{R}^N, \mathbf{y} \in \mathbb{R}^M,$$

where $N$ and $M$ are the dimensions of $\theta$ and $\phi$ respectively. From this, we can derive the following property about the eigenvalue $\lambda = \alpha + i\beta$ with corresponding eigenvector $\mathbf{v} = \mathbf{u}_r + i\mathbf{u}_c$.

$$(H - \lambda)\mathbf{v} = 0 \tag{1}$$
$$\Rightarrow (H - \alpha - i\beta)(\mathbf{u}_r + i\mathbf{u}_c) = 0 \tag{2}$$

Thus the following conditions hold:

$$(H - \alpha)\mathbf{u}_r + \beta\mathbf{u}_c = 0 \tag{3}$$
$$(H - \alpha)\mathbf{u}_c - \beta\mathbf{u}_r = 0. \tag{4}$$

Thus we can multiply the top equation by $\mathbf{u}_r^T$ and the bottom equation by $\mathbf{u}_c^T$ to retrieve the following:

$$\alpha = \frac{\mathbf{u}_r^T H \mathbf{u}_r + \mathbf{u}_c^T H \mathbf{u}_c}{\mathbf{u}_r^T \mathbf{u}_r + \mathbf{u}_c^T \mathbf{u}_c} \geq 0$$

To see that this is strictly above zero, we note that we can split the eigenvector via the following:

$$\mathbf{u}_r + i\mathbf{u}_c = \begin{pmatrix} \mathbf{u}_r^0 + i\mathbf{u}_c^0 \\ \mathbf{u}_r^1 + i\mathbf{u}_c^1 \end{pmatrix}.$$

Thus now $\alpha$ can be rewritten as the following:

$$\alpha = \frac{\left(\mathbf{u}_r^0\right)^T A \mathbf{u}_r^0 + \left(\mathbf{u}_r^1\right)^T C \mathbf{u}_r^1 + \left(\mathbf{u}_c^0\right)^T A \mathbf{u}_c^0 + \left(\mathbf{u}_c^1\right)^T C \mathbf{u}_c^1}{\mathbf{u}_r^T \mathbf{u}_r + \mathbf{u}_c^T \mathbf{u}_c}$$

Since $A \succ 0$, for this to be zero we note that the following must hold

$$\mathbf{u}_r + i\mathbf{u}_c = \begin{pmatrix} \mathbf{0} \\ \mathbf{u}_r^1 + i\mathbf{u}_c^1 \end{pmatrix}.$$

If this is the form of an eigenvector of $H$ then we get the following:

$$\begin{pmatrix} A & B^T \\ -B & C \end{pmatrix} \begin{pmatrix} \mathbf{0} \\ \mathbf{u} \end{pmatrix} = \begin{pmatrix} B^T\mathbf{u} \\ C\mathbf{u} \end{pmatrix} = \lambda \begin{pmatrix} \mathbf{0} \\ \mathbf{u} \end{pmatrix}$$

where $\mathbf{u} = \mathbf{u}_r^1 + i\mathbf{u}_c^1$. One condition which is needed is that $B^T\mathbf{u} = \mathbf{0}$. Another condition needed for this to be an eigenvector is that $\lambda$ is an eigenvalue of $C$. However this would mean that the eigenvalue $\lambda$ is real. If this eigenvalue was thus at 0, the matrix $H$ would not be invertible. Thus concluding the proof. □

**Linear Analysis:** The last part of the proof in Lemma 3.1 can be seen by using linear dynamical systems analysis. Namely when we have the dynamics given by $\dot{x} = Ax$ where $x \in \mathbb{R}^n$ and $A \in \mathbb{R}^n \times \mathbb{R}^n$ is a diagonalisable matrix with eigenvalues $\lambda_1, \cdots \lambda_n$. Then we can decompose $x(t) = \sum_i \alpha_i(t)v_i$ where $v_i$ is the corresponding eigenvector to eigenvalue $\lambda_i$. The solution can be derived as $x(t) = \sum_i \alpha_i(0)e^{\lambda_i t}v_i$.

## B  Off-Diagonal Elements are Opposites at the Nash

Here we show that the off diagonal elements of the Hessian with respect to the Wasserstein, cross-entropy and non-saturating loss are opposites. For the cross-entropy and Wasserstein losses, this property hold for zero-sum games by definition. Thus we only show this for the non-saturating loss.

For the non-saturating loss, the objectives for the discriminator and the generator is given by the following:

$$\ell_D(\theta, \phi) = -\int_x p_d(x) \log(D(x; \theta)) \mathrm{d}x + \int_z p(z) \log(1 - D(G(z; \phi); \theta)) \mathrm{d}z \tag{5}$$

$$\ell_G(\theta, \phi) = -\int_z p(z) \log(D(G(z; \phi); \theta)) \mathrm{d}z. \tag{6}$$

We transform this with $x = G(z; \phi)$ where $x$ is now drawn from the probability distribution $p_g(x; \phi)$. We can rewrite this loss as the following:

$$\ell_D(\theta, \phi) = -\int_x (p_d(x) \log(D(x; \theta)) + p_g(x; \phi) \log(1 - D(x; \theta))\mathrm{d}x \qquad (7)$$

$$\ell_G(\theta, \phi) = -\int_x p_g(z; \phi) \log(D(x; \theta))\mathrm{d}x. \qquad (8)$$

Note that $\frac{\partial^2 \ell_D}{\partial \phi \partial \theta} = -\frac{\partial^2 \ell_G}{\partial \theta \partial \phi}^T$ if and only if the following condition is true

$$\frac{\partial^2 (\ell_D + \ell_G)}{\partial \theta \partial \phi} = 0.$$

For the non-saturating loss this becomes the following:

$$\frac{\partial^2 (\ell_D + \ell_G)}{\partial \theta \partial \phi} = -\int_x \frac{\partial D(x; \theta)}{\partial \theta} \frac{\partial p_g(x; \phi)}{\partial \phi} \left( \frac{1}{D(x; \theta)} - \frac{1}{1 - D(x; \theta)} \right) \mathrm{d}x \qquad (9)$$

At the global Nash, we know that $D(x) = p_d(x)/(p_d(x) + p_g(x; \phi))$ and $p_g(x; \phi) = p_d(x)$, thus this is identically zero.

## C Hessian with Respect to the Discrimator's Parameters is Semi-Positive Definite for Piecewise-Linear Activation Functions

In this section, we show that when we use piecewise-linear activation functions such as ReLU or LeakyReLUs in the case of the *cross-entropy loss* or the *non saturating loss* (as they are the same loss for the discriminator), the Hessian wrt. the discriminator's parameters will be semi-positive definite. Here we also make the assumption that we are never at the point where the piece-wise function switches state (as the curvature is not defined at these points). Thus we note

$$\frac{\partial^2 D}{\partial \theta^2} = 0.$$

For the cross-entropy loss and non-saturating losses, the discriminator network often outputs the logit, $D(x; \theta)$, for a sigmoid function $\sigma(x) = 1/(1 + e^{-x})$. The Hessian with respect to the parameters is given by:

$$\ell_D(\theta, \phi) = -\int_x p_D(x) \log(\sigma(D(x; \theta))) + p_G(x) \log(1 - \sigma(D(x; \theta)))\mathrm{d}x$$

$$\frac{\partial \ell_D}{\partial \theta} = -\int_x (p_D(x)(1 - \sigma(D(x; \theta))) - p_G(x)\sigma(D(x; \theta))) \frac{\partial D(x; \theta)}{\partial \theta}\mathrm{d}x$$

$$\Rightarrow \frac{\partial^2 \ell_D}{\partial \theta^2} = \int_x (p_D(x) + p_G(x)) \sigma(D(x; \theta))(1 - \sigma(D(x; \theta))) \frac{\partial D(x; \theta)}{\partial \theta} \frac{\partial D(x; \theta)}{\partial \theta}^T \mathrm{d}x \succeq 0. \qquad (10)$$

## D Gradient Norm Regularisation and Effects on Integration Error

Here we provide intuition for why gradient regularisation after applying the `ODEStep` might be needed. We start by considering how we can approximate the truncation error of Euler's method with stepsize $h$.

The update rule of Euler is given by

$$y_{k+1} = y_k + h\mathbf{v}(y_k).$$

Note here $y_k = (\theta_k, \phi_k)$. The truncation error is approximated by comparing this update to that when we half the step size and take two update steps:

$$\tilde{y}_{k+1} = y_k + \frac{h}{2}\mathbf{v}(y_k) + \frac{h}{2}\mathbf{v}\left(y_k + \frac{h}{2}\mathbf{v}(y_k)\right) \tag{11}$$

$$= y_k + h\mathbf{v}(y_k) + \frac{h^2}{4}\mathbf{v}(y_k)\frac{\partial \mathbf{v}}{\partial y}(y_k) + O(h^3). \tag{12}$$

$$= y_{k+1} + \frac{h^2}{4}\mathbf{v}(y_k)\frac{\partial \mathbf{v}}{\partial y}(y_k) + O(h^3) \tag{13}$$

$$\Rightarrow \tau_{k+1} = \tilde{y}_{k+1} - y_{k+1} \sim \frac{h^2}{4}\mathbf{v}(y_k)\frac{\partial \mathbf{v}}{\partial y}(y_k) = O(|\mathbf{v}|). \tag{14}$$

Here we see that the truncation error is linear with respect to the magnitude of the gradient. Thus we need the magnitude of the gradient to be bounded in order for the truncation error to be small.

## E  Numerical Integration Update Steps

**Euler's Method:**

$$\begin{pmatrix} \theta_{k+1} \\ \phi_{k+1} \end{pmatrix} = \begin{pmatrix} \theta_k \\ \phi_k \end{pmatrix} + h\mathbf{v}(\theta_k, \phi_k) \tag{15}$$

**Heun's Method (RK2):**

$$\begin{pmatrix} \tilde{\theta}_k \\ \tilde{\phi}_k \end{pmatrix} = \begin{pmatrix} \theta_k \\ \phi_k \end{pmatrix} + h\mathbf{v}(\theta_k, \phi_k)$$

$$\begin{pmatrix} \theta_{k+1} \\ \phi_{k+1} \end{pmatrix} = \begin{pmatrix} \theta_k \\ \phi_k \end{pmatrix} + \frac{h}{2}\left(\mathbf{v}(\theta_k, \phi_k) + \mathbf{v}(\tilde{\theta}_k, \tilde{\phi}_k)\right) \tag{16}$$

**Runge Kutta 4 (RK4):**

$$\mathbf{v}_1 = \mathbf{v}(\theta_k, \phi_k)$$

$$\mathbf{v}_2 = \mathbf{v}\left(\theta_k + \frac{h}{2}(\mathbf{v}_1)_\theta, \phi_k + \frac{h}{2}(\mathbf{v}_1)_\phi\right)$$

$$\mathbf{v}_3 = \mathbf{v}\left(\theta_k + \frac{h}{2}(\mathbf{v}_2)_\theta, \phi_k + \frac{h}{2}(\mathbf{v}_2)_\phi\right)$$

$$\mathbf{v}_4 = \mathbf{v}(\theta_k + h(\mathbf{v}_3)_\theta, \phi_k + h(\mathbf{v}_3)_\phi)$$

$$\begin{pmatrix} \theta_{k+1} \\ \phi_{k+1} \end{pmatrix} = \begin{pmatrix} \theta_k \\ \phi_k \end{pmatrix} + \frac{h}{6}\left(\mathbf{v}_1 + 2\mathbf{v}_2 + 2\mathbf{v}_3 + \mathbf{v}_4\right). \tag{17}$$

Note $\mathbf{v}_\theta$ corresponds to the vector field element corresponding to $\theta$, similarly for $\mathbf{v}_\phi$.

## F  Existing Methods which Approximates Second-order ODE Solver

Here we show that previous methods such as consensus optimisation [8], SGA [1] or Crossing-the-Curl [4], and extragradient methods [7, 2] approximates second-order ODE solvers[1].

To see this let's consider an "second-order" ODE solver of the following form:

$$\begin{pmatrix} \tilde{\theta}_k \\ \tilde{\phi}_k \end{pmatrix} = \begin{pmatrix} \theta_k \\ \phi_k \end{pmatrix} + \gamma\mathbf{v}(\theta_k, \phi_k) = \begin{pmatrix} \theta_k \\ \phi_k \end{pmatrix} + \mathbf{h} \tag{18}$$

$$\begin{pmatrix} \theta_{k+1} \\ \phi_{k+1} \end{pmatrix} = \begin{pmatrix} \theta_k \\ \phi_k \end{pmatrix} + \frac{h}{2}\left(a\mathbf{v}(\theta_k, \phi_k) + b\mathbf{v}(\tilde{\theta}_k, \tilde{\phi}_k)\right) \tag{19}$$

where $\mathbf{h} = \gamma[\mathbf{v}_\theta, \mathbf{v}_\phi]$ and $a, b$ scales of each term.

**Extra-gradient:** Note that when $a = 0$ and $b = 2$ and $\gamma = h$, this is the extra-gradient method by definition [7].

**Consensus Optimisation:** Note that $a = b = 1$ and $\gamma = h$, we get Heun's method (RK2). For consensus optimisation, we only need $a = b = 1$. If we perform Taylor Expansion on Eq (19):

$$\begin{pmatrix} \theta_{k+1} \\ \phi_{k+1} \end{pmatrix} = \begin{pmatrix} \theta_k \\ \phi_k \end{pmatrix} + h\mathbf{v}(\theta_k, \phi_k) + h\left((\nabla\mathbf{v})^T\mathbf{h} + O(|\mathbf{h}|^2)\right). \tag{20}$$

With this, we see that this approximates consensus optimisation [8].

**SGA/Crossing the Curl:** To see we can approximate one part of SGA [1]/Crossing-the-Curl [4], the update is now given by:

$$\begin{pmatrix} \theta_{k+1} \\ \phi_{k+1} \end{pmatrix} = \begin{pmatrix} \theta_k \\ \phi_k \end{pmatrix} + \frac{h}{2}\left(\mathbf{v}(\theta_k, \phi_k) + \begin{pmatrix} \mathbf{v}_\theta(\theta_k, \tilde{\phi}_k) \\ \mathbf{v}_\phi(\tilde{\theta}_k, \phi_k) \end{pmatrix}\right) \tag{21}$$

where $\mathbf{v}_\theta$ denotes the element of the vector field corresponding to $\theta$ and similarly for $\mathbf{v}_\phi$, and $\tilde{\theta}_k, \tilde{\phi}_k$ are given in Eq. (18). The Taylor expansion of this has only the off-diagonal block elements of the matrix $\nabla\mathbf{v}$. More explicitly this is written out as the following:

$$\begin{pmatrix} \theta_{k+1} \\ \phi_{k+1} \end{pmatrix} = \begin{pmatrix} \theta_k \\ \phi_k \end{pmatrix} + \frac{h}{2}\begin{pmatrix} \mathbf{v}_\theta + \gamma\mathbf{v}_\phi\frac{\partial\mathbf{v}_\theta}{\partial\phi} \\ \mathbf{v}_\phi + \gamma\mathbf{v}_\theta\frac{\partial\mathbf{v}_\phi}{\partial\theta} \end{pmatrix} + O(h|\mathbf{h}|^2) \tag{22}$$

All algorithms are known to improve GAN's convergence, and we hypothesise that these effects are also related to improving numerical integration.

## G    Experimental Setup

We use two GAN architectures for image generation of CIFAR-10: the DCGAN [11] modified by Miyato et al. [10] and the ResNet [6] from Gulrajani et al. [5], with additional parameters from [10], but removed spectral-normalisation. For conditional ImageNet $128 \times 128$ generation we use a similar ResNet architecture from Miyato et al. [10] with conditional batch-normalisation [3] and projection [9] but with spectral-normalisation removed. We also consider another ResNet where the size of the hidden layers is $2\times$ the previous ResNet, we also increase the latent size from 128 to 256, we denote this as ResNet (large).

### G.1    Hyperparameters for CIFAR-10 (DC-GAN)

For Euler, RK2 and RK4 integration, we first set $h = 0.01$ for 500 steps. Then we go to $h = 0.04$ till 400k steps and then we decrease the learning rate by half. For Euler integration we found that $h = 0.04$ will be unstable, so we use $h = 0.02$. We train using batch size 64. The regularisation weight used is $\lambda = 0.01$. For the Adam optimiser, we use $h_G = 1 \times 10^{-4}$ for the generator and $h_D = 2 \times 10^{-4}$ for the discriminator, with $\lambda = 0.1$.

### G.2    Hyperparameters for CIFAR-10 (ResNet)

For RK4 first we set $h = 0.005$ for the first 500 steps. Then we go to $h = 0.01$ till 400k steps and then we decrease the learning rate by $0.5$. We train with batch size 64. The regularisation weight used is $\lambda = 0.002$. For the Adam optimiser, we use $h = 2 \times 10^{-4}$ for both the generator and the discriminator, with $\lambda = 0.1$.

### G.3    Hyperparameters for ImageNet (ResNet)

The same hyperparameters are used for ResNet and ResNet (large). For RK4 first we set $h = 0.01$ for the first 15k steps, then we go to $h = 0.02$. We train with batch size 256. The regularisation weight used is $\lambda = 0.00002$.

# H  Additional Experiments

Here we show experiments on ImageNet; ablation studies on using different orders of numerical integrators; effects of gradient regularisation; as well as experiments on combining the RK4 ODE solver with the Adam optimiser.

## H.1  Conditional ImageNet Generation

Figure H1: Comparison of IS and FID for ODE-GAN (RK4) vs. SNGAN trained on ImageNet $128 \times 128$. In the plot on the left we used a ResNet architecture similar to Miyato et al. [10] for ImageNet; on the right we trained with ResNet (large).

Table H1: Comparison of ODE-GAN and the SN-GAN trained on ImageNet (*conditional* image generation) on ResNet (large).

|          | FID (best) / FID (final) | IS (best) / IS (final) |
|----------|--------------------------|------------------------|
| SN-GAN   | 45.62 / 70.38            | 24.64 / 16.01          |
| ODE-GAN  | **22.29 / 23**.46        | **46.17 / 44.66**      |

Fig. H1 shows that ODE-GAN can significantly improve upon SNGAN with respect to both Inception Score (IS) and Fréchet Inception Distance (FID). As in [10] we use conditional batch-normalisation [3] and projection [9]. Similarly to what we have observed in CIFAR-10, the performance degrades over the course of training when we use SNGAN whereas ODE-GAN continues improving. We also note that as we increase the architectural size (i.e. increasing the size of hidden layers $2\times$ and latent size to 256) the IS and FID we obtain using SNGAN gets worse. Whereas, for ODE-GAN we see an improvement in both IS and FID, see Table H1 and Fig. H1. We want to note that our algorithm seems to be more prone to landing in NaNs during training for conditional models, something we would like to further understand in future work.

## H.2  Ablation Studies

Table H2: Ablation Studies for ODE-GAN for CIFAR-10 (DCGAN).

| Solver | $\lambda$ | FID (best) / FID(final) | IS (best)/IS(final) |
|--------|-----------|-------------------------|---------------------|
| | | **ODE Solvers** | |
| Euler | 0.01 | $19.43 \pm 0.03$ / $19.76 \pm 0.02$ | $7.85 \pm 0.03$ / $7.75 \pm 0.03$ |
| RK2 | 0.01 | $18.66 \pm 0.24$ / $18.93 \pm 0.33$ | $7.90 \pm 0.01$ / $7.82 \pm 0.00$ |
| RK4 | 0.01 | $\mathbf{17.66 \pm 0.38}$ / $\mathbf{18.05 \pm 0.53}$ | $\mathbf{7.97 \pm 0.03}$ / $\mathbf{7.86 \pm 0.09}$ |
| | | **Regularisation** | |
| RK4 | 0.001 | $25.09 \pm 0.61$ / $26.93 \pm 0.13$ | $7.27 \pm 0.07$ / $7.17 \pm 0.03$ |
| RK4 | 0.005 | $18.34 \pm 0.24$ / $18.61 \pm 0.30$ | $7.82 \pm 0.03$ / $7.79 \pm 0.06$ |
| RK4 | 0.01 | $\mathbf{17.66 \pm 0.38}$ / $\mathbf{18.05 \pm 0.53}$ | $\mathbf{7.97 \pm 0.03}$ / $\mathbf{7.86 \pm 0.09}$ |
| | | **Adam Optimisation** | |
| Adam | 0.01 | $21.17 \pm 0.10$ / $23.21 \pm 0.25$ | $7.78 \pm 0.08$ / $7.51 \pm 0.05$ |
| RK4 + Adam | 0.01 | $20.45 \pm 0.35$ / $27.74 \pm 1.15$ | $7.80 \pm 0.07$ / $7.18 \pm 0.23$ |
| Adam | 0.1 | $17.90 \pm 0.05$ / $21.88 \pm 0.05$ | $\mathbf{8.01 \pm 0.05}$ / $7.58 \pm 0.06$ |
| RK4 + Adam | 0.1 | $\mathbf{17.47 \pm 0.30}$ / $\underline{23.20 \pm 0.95}$ | $\mathbf{8.00 \pm 0.06}$ / $\underline{7.59 \pm 0.14}$ |

Figure H2: Using RK4 with $\lambda = 0.001$, $h = 0.03$. We show the progression at 0, 2k, 4k and 6k steps. The right most figure shows the losses and the norms of the gradients. The dashed lines are the Nash equilibrium values ($\log(4)$ for the discriminator $\log(2)$ for the generator).

## H.3 Supplementary: Effects of Regularisation

Gradient regularisation allows us to control the magnitude of the gradient. We hypothesise that this helps us control for integration errors. Holding the step size $h$ constant, we observe that decreasing the regularisation weight $\lambda$ lead to increased gradient norms and integration errors (Fig. H4), causing divergence. This is shown explicitly in Fig. H2, where we show that, with low regularisation weight $\lambda$, the losses for the discriminator and the generator start oscillating heavily around the point where the gradient norm rapidly increases.

Figure H3: Comparison between SN vs gradient regularisation (without `ODEStep`). Both are trained with Adam optimisation, $\lambda = 0.01$.

Figure H4: We plot the truncation error over time for different $\lambda$ (shown in legend) and $h = 0.04$.

**Gradient Regularisation vs SN and Truncation Error Analysis** We find that our regularisation (Grad Reg) outperforms spectral normalisation (SN) as measured by FID and IS (Fig. H3). Meanwhile, Fig. H4 depicts the integration error (from Fehlberg's method) over the course of training. As is visible, heavier regularisation leads to smaller integration errors.

# I ODE-GAN Samples

Figure I5: Architecture used DCGAN (CIFAR-10). Sample of images generated using ODE-GAN with $h = 0.04$, RK4 as the integrator and $\lambda = 0.01$.

Figure I6: Architecture used ResNet (CIFAR-10). Sample of images generated using ODE-GAN (RK4) with $h = 0.01$, RK4 as the integrator and $\lambda = 0.01$.

Figure I7: Architecture used ResNet (ImageNet conditional). Sample of images generated using ODE-GAN (RK4) with $h = 0.02$, RK4 as the integrator and $\lambda = 0.00002$.

## Footnotes

[1]Note that methods such as consensus optimisation or SGA/Crossing-the-Curl are methods which address the rotational aspects of the gradient vector field.