[Reviews · NeurIPS 2020]

Review 1

Summary and Contributions: The paper studies the continuous-time dynamics of GAN training and analyzes its behavior in the vicinity of a local Nash equilibrium. Theoretical results show that following the dynamics exactly converges to the local Nash equilibrium and the rate of convergence is independent of the frequency of rotation of the velocity field. The authors further propose solving the dynamics using higher-order ODE solvers such as Runge-Kutta (RK)-2 and RK-4 and experimentally verify that the proposed method works well, notably in the absence of adaptive optimization. An alternative justification for gradient regularization based on the truncation error of the ODE integration is also offered. Experiments have been presented on the CIFAR10 dataset where the proposed method outperforms state-of-the-arts techniques while providing stable optimization.

Strengths: The paper is a significant contribution to the theory of GAN convergence and GAN research in general. Armed with strong theoretical justifications and empirical validation, this work challenges commonly held notions about the (in)stability of GAN training dynamics and the optimization techniques required for training. The empirical evaluation is very well-done. The effect of different optimization techniques, the order of ODE solvers, gradient regularization, etc. has been thoroughly analyzed. Experiments suggest that techniques for Lipschitz regularization such as spectral normalization may perhaps not be necessary for stable GAN training and may even degrade performance in the later phase of optimization. This problem has been alleviated to a large extent by the proposed method. The paper is also an interesting contribution to the evolving body of works connecting neural networks with dynamical systems. In summary, the paper puts forth some exciting ideas and theory that may be relevant to the wider community of generative modeling researchers.

Weaknesses: The paper does not suffer from any major weakness. My only complaint is that the empirical evaluation has been conducted on only one real-world dataset (CIFAR10). The addition of more experiments would significantly improve the paper.

Correctness: The claims and methods presented in the paper are correct and have been largely validated by the experiments.

Clarity: The manuscript is very well-written. I have some suggestions that may help improve the clarity for readers. * A brief discussion of differential Nash equilibrium. * A brief discussion of the problems (and the related work) associated with strong rotational vector fields. * A discussion of the rate of convergence in Lemma 3.1 (Line 112) or a reference at least. * Some typos/grammatical errors: ** Line 50: "max min" —> "min max". ** Line 186-187: "Since ... equilibrium." ** Line 190: () used instead of ||. ** Supplementary, Lemma A.2., Line 15: "positive part" —> "real part".

Relation to Prior Work: The related work is discussed sufficiently and the authors clearly explain the similarities and differences with prior work.

Reproducibility: Yes

Additional Feedback: I suggest that the authors add more real-world experiments to further improve this work. Do the authors have any intuitions about why the generator's gradient explodes (line 179)? [Post Rebuttal] The authors have addressed my concerns and have promised to include more experiments in the paper. Their preliminary experiments look satisfactory. I am increasing my score to 8.


Review 2

Summary and Contributions: In this work, the authors aim to study the continuous-time dynamic induced by GAN training. Under mild assumptions (and with commonly GAN loss, i.e., the min-max cross-entropy loss and non-saturating loss), the authors prove that the dynamics are stable and should converge in the vicinity of a differential Nash equilibrium. Therefore, the authors hypothesize that the instability in training GANs arises from the discretization of the continuous dynamics, and point out that accurate integration would improve training stability. Therefore, an ODE-GAN is proposed. Combined with a regularizer on the gradients (aiming to control the integration error), the authors show that ODE solvers can stabilize the GAN training. A higher-order ODE solver generally leads to better convergence.

Strengths: 1) In my opinion, the perspective of forming GAN training as solving ODEs is an interesting idea. Although previous works have analyzed GANs from the perspective of dynamical systems, this work presents a novel view of stabilizing the GAN training. 2) Besides the theoretical analysis, the submission presents comprehensive experiments and related discussions. Although the training was only performed on data from a mixture of Gaussians and CIFAR-10, the authors show the superior of the proposed ODE-GAN over existing GANs. 3) This paper is well-written. The relation between this submission with previous works is well discussed.

Weaknesses: 1) Both the theoretical analysis and experimental validation focus on two kinds of GAN losses, i.e., the min-max cross-entropy loss and the non-saturating loss, I wondering to know the performance of ODE-GAN using other GAN losses. 2) In this work, the proposed ODE-GAN demonstrates better performance than SN-GAN. Yet, in the paper of SN-GAN, the performance of spectral normalization has been proved by training on the ImageNet dataset. If the authors could valid the ODE-GAN on more complicated generation tasks, the score will be further increased.

Correctness: Yes, I think they are correct.

Clarity: Overall, this paper is well written and easy to follow.

Relation to Prior Work: Yes, it is clearly discussed.

Reproducibility: Yes

Additional Feedback: [After Rebuttal] The authors' feedback basically addressed my concerns. Overall, I incline towards acceptance.


Review 3

Summary and Contributions: This paper proposes an explanation for the instability of GAN training, and argues that it comes from the discretization error of a continuous dynamics, since the latter enjoys local stability under some reasonable assumptions. Following this hypothesis, the authors propose to replace standard GAN training (now viewed as a forward Euler discretization of a continuous flow) by a higher order integrator step. It is demonstrated that this, together with a gradient penalizing method, improves the training stability of GAN training.

Strengths: The hypothesis that the instability that arises from GAN training may be due to truncation errors of an otherwise stable dynamical system is interesting. The proposed modification to GAN training is simple, and appears to be effective.

Weaknesses: The hypothesis as it stands now is somewhat under-substantiated. Concretely: From a numerical analysis point of view, truncation error order and long-time convergence of the discrete sequence from numerical differencing are separate concepts. RK methods have higher order convergence on fixed time intervals, but its domain of absolute stability is not fundamentally different from that of forward Euler. All explicit methods suffer from limited stability, especially for stiff or conservative systems. Figure 1 shows this effect, but if one takes smaller step sizes the Euler method will converge. On the contrary, the RK method also has a finite A-stable region here. They just happen to be different in this case, so one can find a step size for which Euler doesn’t converge but RK does. For example, if one takes epsilon=0 in Eq (7), both RK and Forward Euler will diverge – one will need a symplectic integrator in this case. Hence, this example seems misleading to me. To summarize, the instability the authors appears to show may not be due to order of convergence (a finite-time effect), but rather may be due to a long time effect (absolute stability), and the choice of step sizes. Therefore, I feel it is misleading to say that such instabilities are fixed by higher order schemes – they are usually only fixed by implicit or symplectic schemes. One suggestion will be to track down whether it is accuracy or stability of the integrator that affects GAN training – instead of using all explicit schemes, the authors can consider testing implicit schemes or linear multi-step methods. I believe such an analysis will make the central claim in this paper stronger. With these said, I like the approach in this paper, especially the simplicity of the modification, but I hope that the numerical analysis aspect can be made more solid.

Correctness: The analysis appears correct although I have not checked line by line. The experimental design seems sound, although some ablation studies with different schemes I mentioned above may be useful.

Clarity: The paper is written very clearly.

Relation to Prior Work: The relation to previous GAN training work has been discussed in some capacity. I am not very familiar with the latest work in this area, however, so I cannot confidently comment on the novelty of the algorithm.

Reproducibility: No

Additional Feedback: [After rebuttal] I have read the authors' rebuttal. I think it will be good to clarify the identified issues in a revision. The work will be improved if the reason(s) of improvement is more carefully tracked down. --- I list some additional comments here: 1. Line 38 “We find that the main ingredient we need for stable integration is to avoid large integration errors”, this needs rephrasing, and appears to be a tautology, unstable integration will have large errors. If the asymptotic error is controlled the integration will be stable. 2. Line 76 “This is also known as infinitesimal gradient descent” Perhaps this is an inherited term but this appears strange, since (4) is not a gradient flow of a potential function. It is in general a non-gradient dynamics. 3. Numerical implementation: I tried but could not find the code that reproduces the experiments. In particular, I’m wondering if a ready-made RK solver is used, or a plain one is written from scratch? As far as I know, many available RK solvers actually use 2 different orders (e.g. RK23 or RK45) that implements adaptive stepsizes. I assume that the authors have turned this feature off since this will be adaptive.


Review 4

Summary and Contributions: This paper is about the training instability of GANs. The authors argue that the instability is due to the discrete updates in GANs, and they claim that the training can be stabilized by using higher-order ODE solvers. They show that their method is more stable than the other state-of-the-art methods, and it can even be trained to competitive levels without any adaptive optimiser. The main contributions of this paper are as follows: 1. they propose a novel and practical view that frames GAN training as solving ODEs. 2. they design a regulariser on the gradients to improve numerical integration of the ODE. 3. they show that higher-order ODE solvers lead to better convergence for GANs. The Runge-Kutta ODE solver can train GANs to competitive levels without any adaptive optimiser (e.g., Adam).

Strengths: The proposed method is intuitive and practical, and the results show that the method is effective. It's really important to have a stable training method, so that the training process is stable and the generated samples are stable.

Weaknesses: The paper is about training instability of GANs, and it presents a practical solution. However, I have some questions: 1. The authors claim that the training can be stabilized by using higher-order ODE solvers, but would that introduce computational concerns when dealing with large dataset like ImageNet? I would expect to see some analysis/results/comparison about what the additional computational cost for the improved stability. 2. The experiments are mainly focused on the training stability. It would be better to have some quantitative evaluation on the quality of the generated samples in CIFAR-10. 3. Can this be applied to other GAN architectures?

Correctness: Correct

Clarity: The overall writing is clear, and the paper is easy to read.

Relation to Prior Work: Yes

Reproducibility: Yes

Additional Feedback: I think this is a nice paper. The experimental results are impressive, however it's hard to say that the proposed method outperforms the state-of-the-art methods because the state-of-the-art methods are not mentioned in the paper. It would be helpful if the authors could give more details about the baselines and compare the results. I have read the authors response and other reviewers' comments.

[Author Response · NeurIPS 2020]

We would like to thank all the reviewers for their reviews with insightful comments and suggestions. We would kindly ask the reviewers to consider raising the scores, if we have addressed the scalability concern satisfactorily.

**Scalability and ImageNet:** We want to first address concerns regarding scalability. *The overall wall-clock training time for ODE-GAN is comparable with SN-GAN.* While the RK4 algorithm requires four gradient calculations and the regularizer two backward passes, it avoids the extra discriminator updates (5 discriminator updates per generator update in SN-GAN). The ODE update of our algorithm has the same memory footprint as gradient descent, as each gradient in the RK4 step is computed sequentially. Therefore, the scalability of our algorithm is comparable to SN-GAN.

Table 1: Comparison of ODE-GAN and the SN-GAN trained on ImageNet (*unconditional* image generation).

|  | FID | IS |
| --- | --- | --- |
| SN-GAN | $57.24 \pm 0.79$ | $13.83 \pm 0.19$ |
| ODE-GAN | $\mathbf{49.82 \pm 0.93}$ | $\mathbf{15.08 \pm 0.07}$ |

Table 1 summarises additional experiments on ImageNet at resolution 128 x 128 for unconditional image generation. We followed the setup of [1], using the same ResNet and batch size 64 (learning rate $0.01$, regularisation weight $0.001$, all other settings are the same as for CIFAR10). The scores were computed from 10,000 samples. The advantage of our algorithm is already clear from these preliminary results; we are working on improving them further for the final version.

**Reviewer 1:** We agree with your suggestion to include additional context in the manuscript. In particular, we will include an exposition on differential Nash equilibria and rotational fields in the supplementary. Background on the convergence of linear dynamical systems will be added to the proof of Lemma 3.1. **Re. Intuitions about the generator's gradient exploding:** We observed large gradients when the generator samples are poor, where the discriminator may perfectly distinguish the generated samples from data. This potentially sharp decision boundary can drive the magnitude of the generator' gradients to infinity. We will provide additional intuition in the revision.

**Reviewer 2:** We addressed your request regarding scalability/more complex generation experiments such as ImageNet in the first section of this rebuttal. **Re. Other GAN losses:** The majority of our analysis can be extended to other smooth and differentiable loss functions including the Wasserstein loss. We will discuss this in more details in the revision.

**Reviewer 3:** We appreciate your in-depth comments which help us further consolidate the links by which we connect GAN training with numerical analysis. The nature of our work can entail a clash of terminologies that are overloaded in these fields, which we want to clarify first. **Re. terminology** We will add a short paragraph to clarify the following: the "instability" we refer to is of training GANs (for example due to mode collapse); "convergence" refers to convergence to either the differential Nash equilibrium or to a fixed point; "gradient descent" in our case means following the gradient of losses for each player in the game; the term "infinitesimal gradient descent" is indeed inherited from [2] which refers to the dynamics when we tend the step size of gradient descent to zero for the two-player game. Further, line 38 should indeed be changed to "We find that the main ingredient we need for stable GAN *training* is to avoid large integration errors" reflecting our hypothesis that *instability in GAN training (at least near a Nash-equilibrium) arises from discretization errors in the underlying ODE*. This hypothesis hinges on the observation that the purely rotational case (namely when $\epsilon = 0$ in Eq. 7) does not apply to GANs near differential Nash equilibrium under mild assumptions (Lemma 3.1 and Remark). Under this assumption we can use higher order numerical integrators as well as first order (such as Euler) with a lower step-size (as also pointed out by the reviewer) to stabilize GAN training. In practice, for large-scale image generation with GANs, decreasing the step-size for Euler integration, slows down training significantly (see Fig. 3 where Euler is stable but convergence is too slow).We will clarify these issues further in the revision. **Re. solver implementation** We required a fast solver implementation in TensorFlow and thus wrote all integrators from scratch. **Re. suggestions** Thanks for your suggestion regarding implicit/symplectic numerical schemes, this is something we want to explore in the future.

**Reviewer 4:** **Re. Quantitative evaluation on the quality of the generated samples** We have quantitatively evaluated samples generated via the Inception Score and FID (Table 1 and H1), which are the main metrics used in the literature. If there are additional metrics the reviewer would suggest we are happy to include them. **Re. Other GAN architectures** Yes, our method is not restricted to a specific GAN architecture, it can be applied to any GAN setup. In the paper we e.g. used feed-forward nets (Section 5.1), convolutional nets, and ResNets (Section 5.2). We will provide further details on the different architectures to make this more clear.

# References

[1] Takeru Miyato, Toshiki Kataoka, Masanori Koyama, and Yuichi Yoshida. Spectral normalization for generative adversarial networks. *arXiv preprint arXiv:1802.05957*, 2018.

[2] Satinder P Singh, Michael J Kearns, and Yishay Mansour. Nash convergence of gradient dynamics in general-sum games. In *UAI*, pages 541–548, 2000.


[Meta-Review · NeurIPS 2020]

The paper introduces a new perspective for explaining instability in GANs training by analyzing the continuous dynamics of the training algorithm. They first show that these dynamics should converge in the vicinity of the Nash equilibrium and then make the hypothesis that instability is due to the discretization of this dynamics. They show that using higher order ODE time integrators for solving the dynamics help stabilizing the training. The paper is clear, the reviewers agree that this brings a new perspective for analyzing and training GANs and that this is a significant contribution to this topic. The theoretical findings are backed up by a nice empirical evaluation and analysis. Recommendation: accept.